# Effects of Vitamin E Intake and Voluntary Wheel Running on Whole-Body and Skeletal Muscle Metabolism in Ovariectomized Mice

**DOI:** 10.3390/nu17060991

**Published:** 2025-03-12

**Authors:** Youngyun Jin, Hee-Jung Yoon, Ki-Woong Park, Hanall Lee, Yuan Tan, Byung-Jun Ryu, Seung-Min Lee, Chae-Eun Cho, Jae-Geun Kim, Nam-Ah Kim, Young-Min Park

**Affiliations:** 1College of Sport Science, Sungkyunkwan University, Suwon 16419, Republic of Korea; player53@skku.edu; 2Division of Health and Kinesiology, Incheon National University, Incheon 22012, Republic of Korea; yoonhj330@gmail.com (H.-J.Y.); kiwoongpark1995@gmail.com (K.-W.P.); hlee2500@inu.ac.kr (H.L.); tanyuan0312@gmail.com (Y.T.); zzbjryu@inu.ac.kr (B.-J.R.); seung3759@inu.ac.kr (S.-M.L.); chaeeun7002@gmail.com (C.-E.C.); 3Sport Science Institute & Health Promotion Center, Incheon National University, Incheon 22012, Republic of Korea; 4Division of Life Sciences, College of Life Sciences and Bioengineering, Incheon National University, Incheon 22012, Republic of Korea; jgkim@inu.ac.kr; 5Department of Pharmacy, College of Pharmacy, Mokpo National University, Muan 58554, Republic of Korea; 6Department of Biomedicine, Health & Life Convergence Sciences, BK21 Four, Biomedical and Healthcare Research Institute, Mokpo National University, Muan 58554, Republic of Korea

**Keywords:** ovariectomy, vitamin E, voluntary wheel running, mitochondrial function, skeletal muscle

## Abstract

**Background/Objectives:** Ovariectomized rodents experience metabolic dysfunction in whole-body and skeletal muscle. A disrupted balance between oxidative stress and antioxidants might exacerbate metabolic dysfunction in ovariectomized rodents. Dietary antioxidants, such as vitamin E intake, before or during exercise would be beneficial by mitigating the exercise-induced increase in oxidative stress in ovariectomized rodents. The purpose of the current study was to investigate the potential effect of vitamin E intake combined with voluntary exercise on whole-body and skeletal muscle metabolism in ovariectomized mice. **Methods:** This study used C57BL/6J wild-type female mice (*n* = 40, 8 weeks old), which were randomly assigned into sham (SHM), ovariectomy (OVX), ovariectomy with exercise (OVXVE), ovariectomy with vitamin E (OVXV), ovariectomy with exercise and vitamin E (OVXVE) groups. Body composition, resting metabolic rate, glucose tolerance, skeletal muscle mitochondrial function, and protein contents were assessed using dual-energy x-ray absorptiometry, indirect calorimetry, glucose tolerance test, O_2_K OROBOROS, and Western blot, respectively. **Results:** The combined treatment of vitamin E and voluntary wheel running did not show a beneficial effect on whole-body metabolism such as fat mass, energy expenditure, and glucose tolerance. However, independent of exercise intervention, vitamin E intake enhanced mitochondrial function, Peroxisome proliferator-activated receptor gamma coactivator-1 alpha (PGC1-a), and adenosine monophosphate-activated protein kinase (AMPK) levels and also reduced oxidative stress in the skeletal muscles of ovariectomized mice. Specifically, in the soleus muscle, vitamin E intake enhanced mitochondrial function and PGC1-a content (*p* < 0.05). In the gastrocnemius muscle, vitamin E intake enhanced PGC1-a and AMPK levels and reduced a marker of oxidative stress (*p* < 0.05). **Conclusions:** Vitamin E, as a potent antioxidant, may play a crucial role in maintaining skeletal muscle health in ovariectomized mice. More studies are necessary to investigate whether this finding is applicable to women.

## 1. Introduction

The period from an irregular menstrual cycle to the last period is called the menopausal transition [1], and physiological changes during menopausal transition are accompanied by a sharp decrease in circulating estrogen levels [2]. A sharp decrease in estrogen may be associated with most of the risk factors of metabolic syndrome and impose physiological stress on menopausal women [3]. Women experience menopausal transition for a relatively short period of one to five years compared with andropause (i.e., one to ten years) [4]. Thus, women have less time to adapt to dramatic physiological changes, and menopause-related diseases can appear, such as metabolic syndrome, osteoporosis, and cardiovascular disease [5,6].

A previous study reported an association between menopause and a 60% increase in the risk of metabolic syndrome [7]. Further, postmenopausal women indicated a higher risk of metabolic syndrome than premenopausal women [8]. Metabolic syndrome has several characteristics, such as abdominal visceral fat accumulation, dyslipidemia, hyperneutrophilia, high blood pressure, insulin resistance, and elevated low-density lipoprotein [9]. Importantly, metabolic syndrome is a significant public health issue because it is associated with conditions such as stroke, ischemic heart disease, polycystic ovary syndrome, non-alcoholic fatty liver disease, dementia, and endometrial cancer [10]. Women live one-third or more of their lives after menopause, and it is necessary to prevent or delay menopause-induced metabolic syndrome through health management.

In the early 2000s, an increasing number of menopausal women received hormone replacement therapy (HRT) to treat or alleviate menopausal symptoms [11,12]. However, previous findings on HRT indicated conflicting results. For instance, some previous studies suggested that HRT may alleviate postmenopausal symptoms and change body composition (i.e., decreased weight and fat mass) [13,14,15], whereas other studies suggested that HRT may have a negative effect on cardiovascular disease risk of postmenopausal women [16,17]. A previous study reported that the group receiving HRT with estrogen and progesterone increased the incidence of cardiovascular disease by 29% compared with the placebo group. Additionally, HRT was reported to increase the risk of deep vein thrombosis, pulmonary embolism, and venous thromboembolism to be twice as high in comparison to the placebo group [18]. Accordingly, the Women’s Health Initiative (WHI) announced the positive correlation between HRT and the incidence of cardiovascular disease and issued warnings about the use of HRT in postmenopausal women [19]. These previous findings and recommendations from the WHI have increased interest in other non-hormone treatment approaches such as preventive and alternative medicine, exercise intervention, etc.

Estrogen regulates various parts of mitochondrial function, including energy production, mitochondrial membrane potential generation, and mitochondrial biogenesis [20,21]. Additionally, estrogen receptors can regulate mitochondrial function through genomic or non-genomic mechanisms, allowing receptors to directly improve the mitochondrial energy production and metabolic regulation capacity. Estrogen receptors are essential for mitochondrial respiratory and antioxidant proteins that may protect against oxidative stress [22,23]. However, postmenopausal women exhibit impaired estrogen production, which may lead to a decreased antioxidant capacity along with an increment in oxidative stress [24,25,26].

Oxidative stress is critically related with the pathological development of numerous degenerative and chronic diseases, such as atherosclerosis and cancer [27,28,29]. Specifically, the production of reactive oxygen species may be associated with biological damage and may potentially contribute to disease initiation and progression [30,31,32]. Vitamin E is an antioxidant reported to reduce the concentration of mitochondrial reactive oxygen species, potentially mitigating mitochondrial oxidative damage [33,34].

In addition to vitamin E supplements, regular physical activity and exercise are recommended to maintain and strengthen the mitochondrial antioxidant system [35]. Regular exercise has been suggested to increase antioxidant enzyme levels, thereby reducing oxidative stress [36,37,38]. However, the increase in antioxidants from regular exercise may not be physiologically proportional to the demands of increased oxidative stress during exercise. Thus, the importance of dietary antioxidants, such as vitamin E, has been increasingly recognized [39,40,41]. While previous studies have demonstrated that either vitamin E intake or exercise can enhance mitochondrial function, research investigating the combined effects of vitamin E and exercise remains limited. Thus, the purpose of this study was to investigate the potential effect of vitamin E intake with regular exercise on whole-body and skeletal muscle metabolism using the ovariectomized mouse, a gold-standard model of human menopause. The hypothesis was that vitamin E intake combined with exercise treatment would favorably affect whole-body metabolism (i.e., fat mass, energy expenditure, glucose tolerance) and alleviate skeletal muscle mitochondrial dysfunction in ovariectomized mice.

## 2. Materials and Methods

### 2.1. Animals

Forty 8-week-old female C57BL/6J wild-type mice were randomly assigned to five groups: (1) sham control (SHM: n = 8), (2) ovariectomy (OVX: n = 8), (3) ovariectomy with exercise (OVXE: n = 8), (4) ovariectomy with vitamin E (OVXV: n = 8), and (5) ovariectomy with combined exercise and vitamin E (OVXVE: n = 8). The environment of the experimental animal was adjusted to a constant light and dark period (12:12 h light/dark), relative humidity (50–80%), and room temperature (22 ± 1 °C). Animals were single-housed and fed rodent chow (standard or vitamin E-supplemented chow; Rodent NIH-41KO; Zeigler Bros Inc., Gardners, PA, USA) and water ad libitum. All animal experiments were approved by the Institutional Animal Care and Use Committee at Incheon National University (INU-ANIM-2021-04, 10 May 2021).

### 2.2. Experimental Design

After the group assignment, body weight, food intake, and wheel running distance were measured every week. The mice were treated for 13 weeks with each assigned intervention for the respective groups (i.e., OVXE, OVXV, and OVXVE groups). Mice in the exercise groups were provided access to voluntary running wheels (wheel diameter, 10 cm; width of cage, 13 cm; length of cage, 23 cm; height of cage, 14.5 cm), and running distance was monitored with a computerized counter. After 11 weeks of intervention, body composition (whole-body fat mass, lean mass) was measured using dual-energy X-ray absorptiometry (DEXA, GE Medical Systems Ultrasound& Primary Care Diagnostics, LLC, Madison, WI, USA). After setting the scan target as a small animal in the DEXA enCORE v18 software program, the fat and lean mass were reported as the average values obtained from three scans for each animal. After 11–12 weeks of intervention, a resting metabolic rate measurement and glucose tolerance test were performed. On the final measurement day, overnight-fasted mice were anesthetized using 2.5% tribromoethanol at 0.01 mL/g. Gastrocnemius and soleus muscles were collected from one leg for the measurements of mitochondrial function, while the same muscles from the other leg were collected, snap frozen, and stored at −80 °C until protein content analysis. The mice in the exercise groups were blocked from using a voluntary running wheel.

### 2.3. Ovariectomy Surgeries

After ~5 min of inhaling 2% isoflurane through an isoflurane inhalation system (RWD, Life Science Co., Oakland, CA, USA), the disappearance of animal movement was confirmed by stimulating the feet to check for any reflex activity. The isoflurane concentration during surgery was maintained at 0.5%. Subsequently, shaving and sterilization were performed at the center of the dorsal surface, followed by bilateral incisions of less than 0.5 cm on the muscle layer. Following ovary removal, acetaminophen 500 mg/kg was administered, and the incision was closed with small-animal wound clips. The animal’s condition was monitored twice a day for one week of recovery. For the SHM group, all the same surgical procedures were performed, but the ovaries were left intact.

### 2.4. Vitamin E Intake

The vitamin E intake group were fed a diet containing vitamin E (10 IU/g, 1% weight of vitamin E to standard chow, DBL Co., Incheon, Republic of Korea) for 13 weeks [42]. The vitamin E group had free access to vitamin E-supplemented chow (Table 1).

### 2.5. Metabolic Monitoring

The resting metabolic rate was assessed through indirect calorimetry (i.e., assessing oxygen consumption and carbon dioxide production) using a metabolic monitoring system for a 72 h period (Harvard Apparatus, Holliston, MA, USA), as previously described [43]. Mice were singly housed in the system and acclimated to the environment for a day. Twelve-hour REE for light and dark cycles was calculated using 30 min of the lowest energy expenditure. During the 72 h, running wheels in the exercise groups were removed from each metabolic cage.

### 2.6. Glucose Tolerance Test

After fasting for 12 h, 5 to 10 µL of blood was collected through tail veins, and the glucose concentration was measured through a glucometer (Accu-chek Performa, Roche Diagnostics, Mannheim, Germany). The concentration of glucose (pg/mL) was measured before and after the intraperitoneal injection of sterile glucose (1.5 g/kg) at 15, 30, 45, 60, and 120 min. The total area under the curve (tAUC) was used to measure the glucose response from the baseline to the 120 min point [43].

### 2.7. Mitochondrial Function

The mitochondrial function of the soleus and gastrocnemius muscle was investigated using the Oroboros Oxygraph O_2_k (Oroboros Instruments, Innsbruck, Austria) with the modified experimental protocol from the previous study [44]. Following the tissue preparation, muscle tissues were incubated in 2 mL of the specialized buffer (Miro5; 0.5 mM EGTA, 3 mM MgCl_2_·6H_2_O, 60 mM K-lactobionate, 20 mM taurine, 10 mM KH_2_PO_4_, 20 mM HEPES, 110 mM sucrose, 1 g/L bovine serum albumin, pH 7.0). Following the adjustment of O_2_ concentration in the chambers to 350~380 nmol/mL using 200 mM H_2_O_2_ and 112,000 U/mL catalase, mitochondrial respiration was assessed in response to the following substrates: 5 mM pyruvate, 0.5 mM malate, 2 mM adenosine 5′-diphosphate (ADP), and 9 mM succinate. Mitochondrial function was analyzed by comparing differences between groups for three administrations (pyruvate and malate, ADP, and succinate).

### 2.8. Western Blot Analysis

SDS-PAGE Western blots were performed, as previously referenced [45]. Frozen whole-soleus and gastrocnemius skeletal muscles were homogenized in the Triton X-100 solution containing protease and phosphatase inhibitors using a tissue homogenizer (TissueLyser, Qiagen, MD, USA). The total protein concentration was assessed using an assay kit (Pierce BCA protein assay, Thermo Fisher Scientific, Waltham, MA, USA). Homogenates mixed in Laemmli buffer were separated by SDS-PAGE and transferred to polyvinylidene difluoride (PVDF) membranes (Amersham, Germany). PVDF membranes were incubated with primary antibodies (1:2000 concentration in 5% BSA, bovine serum albumin, or skimmed milk). Peroxisome proliferator-activated receptor gamma coactivator-1 alpha (PGC1-a) antibody was purchased from Santa Cruz (sc517380; Santa Cruz Biotechnology, Dallas, TX, USA). Total adenosine monophosphate-activated protein kinase alpha(AMPKa), phospho-AMPKa (Thr172 activation site), and anti-rat IgG secondary antibodies were purchased from Cell Signaling (#2532, #2535, #7077, Cell Signaling Technology, Danvers, MA, USA). The measure of protein carbonylation, an oxidative stress marker, was determined using 2,4-dinitrophenylhydrazine (DNPH) with the protein carbonyl assay kit (ab178020, Abcam Inc., Waltham, MA, USA). Individual protein bands were quantified with a densitometer (Bio-Rad, Hercules, CA, USA) and normalized to the GAPDH antibody (#5174, Cell Signaling Technology, Danvers, MA, USA).

### 2.9. Statistical Analysis

For the statistical analysis, all data were analyzed using SPSS 25.0 (SPSS Inc., IBM, Armonk, NY, USA). One-way ANOVAs were used to determine statistical differences in food intake, body weight, fat/lean mass, resting metabolic rate, glucose tolerance test, mitochondrial function, and protein content between groups. A repeated measures ANOVA (group × time) was performed on the dependent variables to estimate differences in wheel running distances between groups. When a significant interaction existed, the LSD post hoc test was used for pairwise comparisons. All of the data are reported as mean ± SE, and an alpha level at 0.05 was set.

## 3. Results

### 3.1. Body Weight

The OVX, OVXE, OVXV, and OVXVE groups had significantly higher body weights than the SHM group during the early period (*p* < 0.01; Figure 1A). For the mid-period (Figure 1A), the analysis revealed that the body weight of all groups was significantly higher than that of SHM (*p* < 0.01), and that of the OVXVE group was significantly higher than that of the OVX group (*p* = 0.022). For the late period (Figure 1A), the analysis demonstrated that the OVX (*p* = 0.022), OVXE (*p* = 0.008), OVXV (*p* < 0.001), and OVXVE groups (*p* < 0.001) had higher values of body weight than the SHM group, and the OVXVE group had a significantly higher value than the OVX group (*p* = 0.025).

### 3.2. Fat and Lean Mass

Fat mass measures (Figure 1B) indicated significant group differences: (1) the OVXV (*p* = 0.017) and OVXVE groups (*p* < 0.001) had a significantly higher fat mass than the SHM group; and (2) OVXVE showed a higher fat mass than the OVX (*p* = 0.008) and OVXE groups (*p* = 0.01). However, the analysis failed to indicate any group difference in lean mass (Figure 1C).

### 3.3. Food Intake and Wheel Running Distance

The ANOVA on food intake failed to indicate any significant effects in the early period. For the mid-period, the analysis indicated that OVX (*p* = 0.001), OVXE (*p* = 0.006), and OVXV (*p* = 0.003) had lower values of food intake in grams compared with the SHM group. Additionally, only the OVXV group showed a lower value of food intake compared with the SHM group (*p* = 0.026) in the late period. The analysis showed no group differences in wheel running distance (*p* = 0.422).

### 3.4. Resting Metabolic Rate

The resting metabolic rate during the light time (sleeping period) failed to indicate any significant difference between groups. In contrast, during the dark time (active period), the analysis revealed that the OVX (*p* = 0.004), OVXE (*p* = 0.009), OVXV (*p* = 0.007), and OVXVE groups (*p* = 0.007) had a lower energy resting metabolic rate than the SHM group (Figure 2).

### 3.5. Glucose Tolerance Test

The analysis of GTTAUC demonstrated a significant group difference (Figure 3): (1) the OVX (*p* = 0.002), OVXV (*p* = 0.003), and OVXVE groups (*p* < 0.001) had significantly higher values of GTTAUC than the SHM group, and (2) the OVXVE group exhibited a higher value of GTTAUC than the OVXE group (*p* = 0.001).

### 3.6. Mitochondrial Function

For the mitochondrial function of the soleus muscle, the analysis of the oxygen consumption rate indicated significant group difference (Figure 4A): (1) the OVXVE group had a higher value of O_2_ flux than the SHM (*p* = 0.033), OVX (*p* = 0.010), and OVXE groups (*p* = 0.016) in the ADP section; (2) the OVXVE group had a higher O_2_ flux value than the SHM group (*p* = 0.011), OVX group (*p* = 0.004), and OVXE group (*p* = 0.016) in the succinate section; and (3) the OVXV group had a higher value of O_2_ flux than the SHM group in the succinate section (*p* = 0.011). However, the analysis failed to indicate any significant group differences in the mitochondrial function of the gastrocnemius muscle (Figure 4B).

### 3.7. Protein Contents of Skeletal Muscle

For the soleus muscle, the protein content of PGC1-a indicated a significant group difference (Figure 5A): (1) the OVXVE group showed higher PGC1a content than the SHM (*p* = 0.020) and OVX groups (*p* = 0.004), and (2) the OVXV group tended to have higher PGC1a content than the OVX group (*p* = 0.055). For the gastrocnemius muscle, the analysis indicated a significant group difference (Figure 5B): (1) the OVXV (*p =* 0.026) and OVXVE groups (*p* = 0.012) showed a higher value of PGC1a content than the OVX group. However, the analysis failed to indicate any significant group differences in the pAMPKa/AMPKa ratio, a marker of AMPK activity, of the soleus muscle (Figure 5C). Additionally, the analysis for gastrocnemius muscle revealed significant group differences (Figure 5D): (1) the OVXV group showed a higher pAMPKa/AMPKa ratio than the SHM group (*p* = 0.008), OVX group (*p =* 0.003), and OVXE group (*p* = 0.013); and (2) the OVXVE group exhibited a higher pAMPKa/AMPKa ratio than the OVX group (*p* = 0.040).

The analysis of carbonyl protein levels, a marker of oxidative stress, failed to indicate any significant group difference in soleus muscle (Figure 6A). However, the analysis showed significant group differences in the gastrocnemius muscle (Figure 6B): (1) the OVXV group had a lower value of carbonyl protein than the SHM group (*p* = 0.046), and (2) the OVXV (*p* = 0.002) and OVXVE groups (*p* = 0.034) indicated a lower carbonyl protein value than the OVX group.

## 4. Discussion

The current study investigated the potential effect of vitamin E intake combined with voluntary exercise on whole-body and skeletal muscle metabolism in ovariectomized mice. Overall, the combined treatment of vitamin E and voluntary wheel running failed to show a beneficial effect on whole-body metabolism such as fat mass, energy expenditure, and glucose tolerance. However, in general, vitamin E intake enhanced mitochondrial function and its related proteins in the skeletal muscles of ovariectomized mice. Specifically, in the soleus skeletal muscle, vitamin E intake enhanced mitochondrial function and PGC1-a content. In the gastrocnemius muscle, vitamin E intake enhanced PGC1-a and AMPK protein contents and reduced the marker of oxidative stress.

In the ovariectomized group, body weight was significantly higher compared to the SHM group across the early (i.e., 1–4 weeks), mid (i.e., 5–8 weeks), and late period (i.e., 9–13 weeks) of the study. These findings are consistent with previous studies suggesting that ovariectomy may influence body weight through hormonal changes [46,47]. Women tend to gain weight within the first year of menopause, with an average increase of approximately 5 kg during the initial 36 months of early menopause [48]. The hypothesis was that vitamin E intake or the combined treatment of vitamin E and exercise would reduce fat mass in ovariectomized mice. However, the current findings show that the OVXV and OVXVE groups exhibited greater fat mass compared to the SHM group, with the OVXVE increasing fat mass to a greater extent. According to a previous report, groups with higher dosages of vitamin E intake showed greater increases in body weight compared to groups receiving lower dosages [49]. Prolonged consumption of vitamin E (i.e., for more than 8 weeks) increased alpha-tocopherol levels in plasma and muscle to a saturated state, and this consistent elevation may contribute to an increase in body weight with accumulated lipids [50,51,52]. Whether the combined treatment of vitamin E and exercise affects the appetite or energy expenditure should be studied more in the ovariectomized model.

All treatment groups exhibited significantly lower energy expenditure compared with the SHM group. The potential degenerative effects of menopause may lead to reduced energy expenditure and consequent fat accumulation [53]. Indeed, postmenopausal women exhibited a 10% lower resting metabolic rate in comparison to premenopausal women [54]. Contrary to the hypothesis of the current study, the current study found that vitamin E or exercise treatment failed to attenuate the ovariectomy-induced decrease in energy expenditure. The current study utilized an indirect calorimetry metabolic chamber to estimate the metabolic activity of mice. However, the chamber could not accommodate running wheels for the exercise groups, meaning that the measured energy expenditure primarily reflects the resting or sedentary state and may not fully represent total energy expenditure.

Ovariectomy can impair glucose processing capability, as the drastic decrease in circulating estrogen levels may reduce glucose uptake and utilization in both muscles by lowering the efficiency of insulin receptor signaling [55]. Another study reported that ovariectomized mice exhibited a decrease in insulin-induced glucose uptake in skeletal muscle [56]. Similarly to these findings, the glucose tolerance test showed impaired blood glucose dynamics following ovariectomy. However, contrary to our hypothesis, the OVXV and OVXVE group, but not the OVXE group, showed impaired glucose dynamics compared to the SHM group. This suggests that ovariectomy impaired glucose tolerance, but exercise treatment partially attenuates this impairment. Also, vitamin E intake and exercise treatment failed to attenuate ovariectomy-induced glucose intolerance. It is well documented that exercise or regular physical activity improves glucose tolerance. Following exercise, the glucose absorption rate increased due to enhanced insulin sensitivity, potentially offsetting the impact of ovarian hormone deficiency [57,58]. Indeed, ovariectomized mice enhanced glucose tolerance with increased insulin sensitivity following exercise training [59]. Another clinical study also reported improvements in both glucose tolerance and insulin sensitivity as a result of regular exercise [60]. More studies are necessary to investigate how vitamin E intake itself or combined with exercise impairs glucose tolerance in ovariectomized mice.

Vitamin E has been considered an antioxidant that reduces the concentration of mitochondrial reactive oxygen species and mitigates consequent mitochondrial oxidative damage [34]. The current study demonstrated that the combined treatment of vitamin E intake and exercise improved mitochondrial function in the soleus muscle such that the OVXVE group showed the greatest oxygen consumption among all treatment groups. This finding was congruent with PGC1-a protein content; specifically, the OVXVE showed the greatest level of PGC1-a among all groups, implying the synergistic effect of the combined treatment of vitamin E and exercise. As the enhanced levels of antioxidants following exercise would not be proportional to the demands of rapidly increased oxidative stress during an exercise bout, the possibility exists that the supplemented dietary antioxidants, such as vitamin E, would be beneficial to mitigating the exercise-induced oxidative stress and consequent mitochondrial dysfunction [40]. Another potential effect of vitamin E intake and exercise on mitochondrial function may involve a compensatory mechanism that helps maintain energy balance by enhancing mitochondrial function. Specifically, gaining weight and positive energy balance may lead to an increase in total energy expenditure, as both resting metabolic rate and spontaneous physical activity-related energy expenditure may rise to maintain energy balance in overweight or obese people [61]. In the current study, enhanced mitochondrial function in the OVXVE group might be associated with their positive energy balance state, as the OVXVE group was the group that showed the greatest increase in body weight and fat mass. However, this theory does not fully comprehend the current findings. Based on the energy balance theory, although the OVXVE group exhibited the greatest body weight and fat mass compared with the other groups, the resting metabolic rate did not differ among groups. Thus, further investigation is needed to elucidate the potential mechanisms by which the combined treatment of vitamin E intake and exercise enhanced mitochondrial function in the soleus muscle.

Overall, the current study demonstrated that, independent of exercise intervention, vitamin E intake enhanced mitochondrial function and its related proteins in both soleus and gastrocnemius muscles. The discrepancy exists between soleus and gastrocnemius in that the soleus showed a significantly enhanced mitochondrial function and PGC1-a in the groups with vitamin E intake, whereas the gastrocnemius showed a positive effect in PGC1-a, AMPK, and oxidative stress marker. The discrepancy between soleus and gastrocnemius would be related to its difference in muscle fiber composition. A higher proportion of Type I muscle fibers was found in the soleus, which provide a more efficient process of oxidative metabolism [62]. Type I fibers contain approximately three times more triglycerides than Type II fibers and exhibit increased catalase activity [62]. Compared to the gastrocnemius, the soleus muscle containing greater Type I fibers has been believed to be the fibers with exceptional adaptation capacity in response to aerobic exercise by increasing enzymes related to oxygen consumption. Importantly, the current study employed voluntary wheel running for mice, a form of exercise primarily associated with a higher facilitation of aerobic and mitochondrial metabolism in the soleus rather than gastrocnemius muscle. However, more studies are needed on the underpinning mechanism by which vitamin E intake increases oxygen consumption (i.e., mitochondrial function) and PGC1-a content without alterations in AMPK and carbonyl protein in the soleus muscle.

It has been documented that vitamin E intake reduced oxidative damage via decreasing carbonyl protein levels in mitochondria [63,64]. In line with previous studies, the current study found that, in gastrocnemius muscle, vitamin E intake has a positive effect on a marker of oxidative stress such as a decrease in carbonyl protein. The activation of AMPK is considered critical for various cellular pathways including those involved in insulin signaling and mitochondrial biogenesis [65], while PGC1-a is related to oxidative metabolism, which facilitates oxidative phosphorylation by increasing the expression of genes encoding enzymes in the mitochondrial respiratory chain [66,67]. The current study found an increase in AMPK and PGC1-a following vitamin E intake in gastrocnemius muscle, suggesting that the decreased levels of carbonyl protein following vitamin E intake may be associated with AMPK and PGC1-a activation. Previous findings indicated that the AMPK activation in skeletal muscle was shown to increase following exercise intervention in ovariectomized mice [59]. The initial hypothesis of the current study was in line with this previous finding. However, no additional effect of exercise on vitamin E intake was found in AMPK and carbonyl protein in the gastrocnemius muscle.

This study has several potential limitations, and the findings should be interpreted with caution. As this study was conducted using ovariectomized mice, extrapolating the results to actual women requires careful consideration. Second, the total energy expenditure of the OVXE and OVXVE groups could not be measured due to insufficient space for a running wheel in the metabolic chamber. Future studies should employ metabolic chambers containing a running wheel. The experiment with a running wheel would be able to obtain the precise difference between groups in whole-body metabolism. Third, a normality test such as the Shapiro–Wilk test should be performed before carrying out ANOVAs. However, since each experimental group has a minimal sample size due to ethical and cost issues, the current study assumed normality. The findings should be interpreted with caution. Lastly, as the current findings focused on physiological changes (i.e., whole-body metabolic rate, mitochondrial function, protein expression, etc.), future studies should investigate the potential effects of menopause-related psychological and behavioral changes (e.g., dietary intake or spontaneous cage activity) on physiological changes. This would provide a clue for solving the problem found in the current study, such as the enhanced body weight and fat mass in the group of combined treatments.

## 5. Conclusions

The current study investigated the potential effect of vitamin E intake combined with regular exercise on whole-body and skeletal muscle metabolism in ovariectomized mice. The combined treatment of vitamin E and voluntary wheel running did not show a beneficial effect on whole-body metabolism such as fat mass, energy expenditure, and glucose tolerance. However, independent of exercise intervention, vitamin E intake enhanced mitochondrial function, PGC1-a, and AMPK proteins and also reduced oxidative stress in the skeletal muscles of ovariectomized mice. More studies are necessary to investigate the underlying mechanisms by which vitamin E intake enhances skeletal muscle mitochondrial function in ovariectomized mice.

## Figures and Tables

**Figure 1 nutrients-17-00991-f001:**
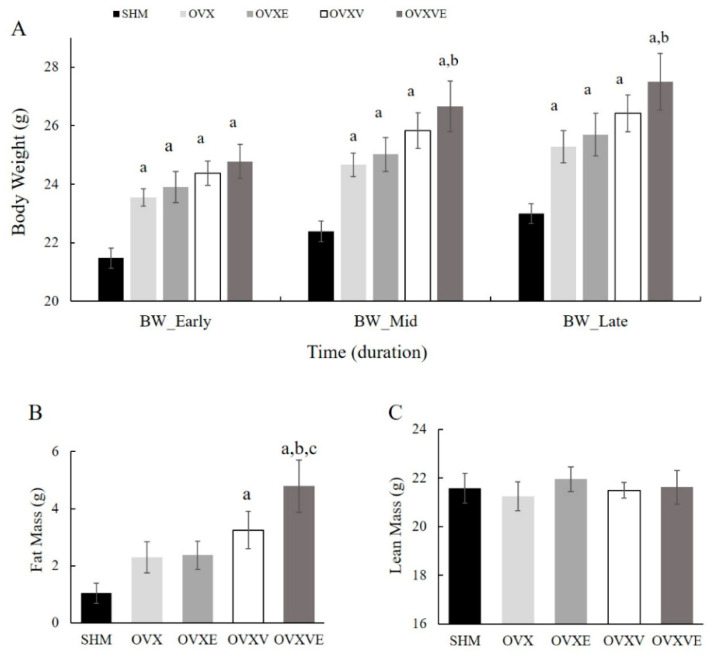
Body composition. (**A**) Changes in body weight between groups per period, (**B**) differences in fat mass between groups, and (**C**) differences in lean mass between groups. Values are means ± SE. ^a^ a significant difference compared to the SHM group; ^b^ a significant difference compared to the OVX group; ^c^ a significant difference compared to the OVXE group. SHM: sham group; OVX: ovariectomy group; OVXE: ovariectomy with exercise; OVXV: ovariectomy with vitamin E; OVXVE: ovariectomy with exercise and vitamin E; BW: body weight.

**Figure 2 nutrients-17-00991-f002:**
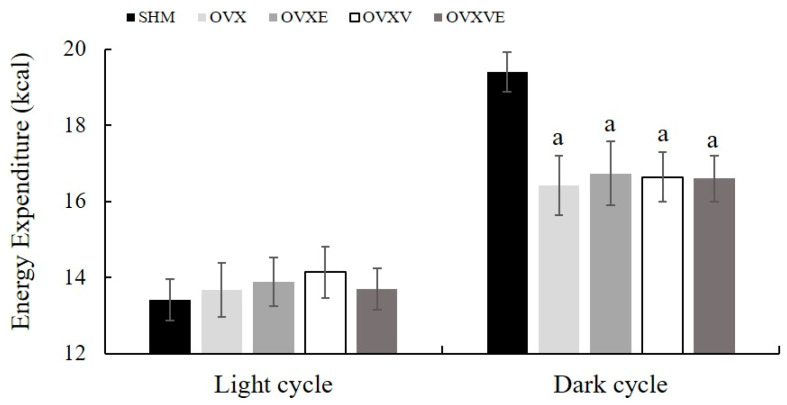
Resting metabolic rate. Energy expenditure during the light and the dark cycle. Values are means ± SE. ^a^ a significant difference compared to the SHM group. SHM: sham group; OVX: ovariectomy group; OVXE: ovariectomy with exercise; OVXV: ovariectomy with vitamin E; OVXVE: ovariectomy with exercise and vitamin E.

**Figure 3 nutrients-17-00991-f003:**
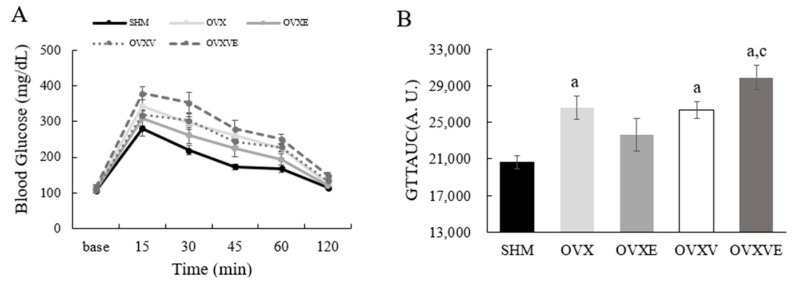
Glucose tolerance test. (**A**) Blood glucose kinetics during glucose tolerance test and (**B**) area under the curve during the glucose tolerance test. Values are means ± SE. ^a^ a significant difference compared to the SHM group; ^c^ a significant difference compared to the OVXE group. SHM: sham group; OVX: ovariectomy group; OVXE: ovariectomy with exercise; OVXV: ovariectomy with vitamin E; OVXVE: ovariectomy with exercise and vitamin E; GTTAUC: glucose tolerance test area under the curve.

**Figure 4 nutrients-17-00991-f004:**
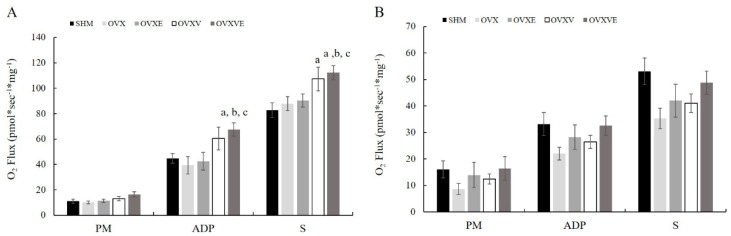
Mitochondrial function in skeletal muscle. (**A**) The oxygen consumption in the soleus and (**B**) gastrocnemius muscle. Values are means ± SE. ^a^ a significant difference compared to the SHM group; ^b^ a significant difference compared to the OVX group; ^c^ a significant difference compared to the OVXE group. SHM: sham group; OVX: ovariectomy group; OVXE: ovariectomy with exercise; OVXV: ovariectomy with vitamin E; OVXVE: ovariectomy with exercise and vitamin E; PM: pyruvate and malate; ADP: adenosine diphosphate; S: succinate.

**Figure 5 nutrients-17-00991-f005:**
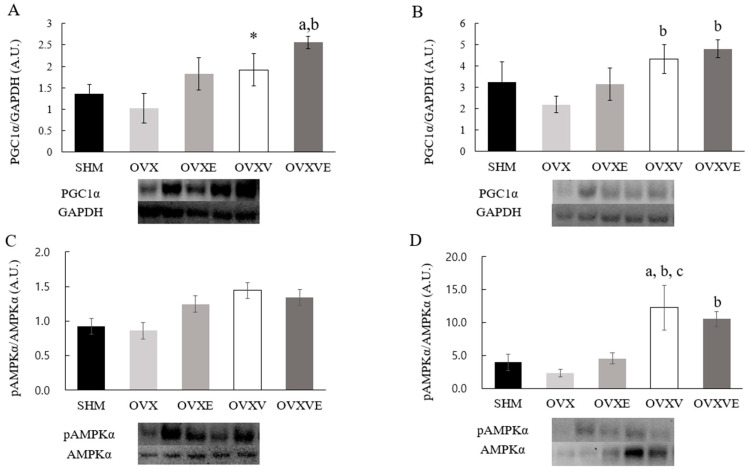
Protein contents in skeletal muscle. (**A**) PGC1a protein content in soleus muscle, (**B**) PGC1a protein content in gastrocnemius muscle, (**C**) AMPKa protein content in soleus muscle, and (**D**) AMPKa protein content in gastrocnemius muscle. Values are means ± SE. ^a^ a significant difference compared to the SHM group; ^b^ a significant difference compared to the OVX group; ^c^ a significant difference compared to the OVXE group; * a trend compared to the OVX group (*p* = 0.055). SHM: sham group; OVX: ovariectomy group; OVXE: ovariectomy with exercise; OVXV: ovariectomy with vitamin E; OVXVE: ovariectomy with exercise and vitamin E.

**Figure 6 nutrients-17-00991-f006:**
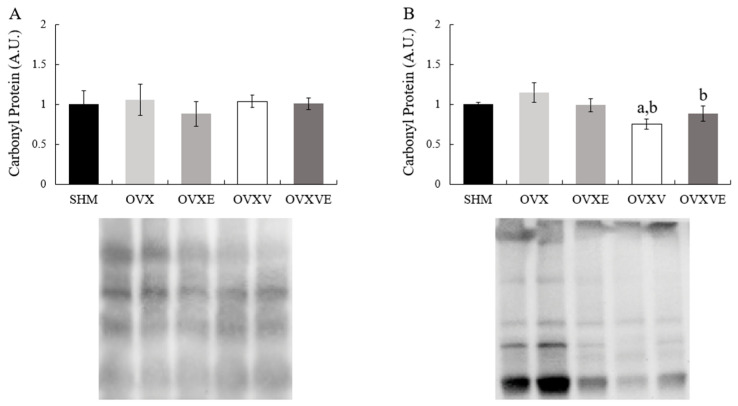
Carbonyl protein contents in skeletal muscle. (**A**) Carbonyl protein contents in soleus muscle and (**B**) carbonyl protein contents in gastrocnemius muscle. Values are means ± SE. ^a^ a significant difference compared to the SHM group; ^b^ a significant difference compared to the OVX group. SHM: sham group; OVX: ovariectomy group; OVXE: ovariectomy with exercise; OVXV: ovariectomy with vitamin E; OVXVE: ovariectomy with exercise and vitamin E.

**Table 1 nutrients-17-00991-t001:** Dietary composition.

Ingredient	CON	VIT
Ground Whole Wheat (%)	34.90	34.55
Ground No.2 Yellow Corn (%)	21.00	20.79
Ground Whole Oats (%)	10.00	9.9
Wheat Middlings (%)	10.00	9.9
Fish Meal (60% protein)	9.00	8.91
Soybean Meal (47.5%)	5.00	4.95
Alfalfa (17% protein)	2.00	1.98
Corn Gluten meal (60% protein)	2.00	1.98
**Major Nutrient**	**CON**	**VIT**
Alpha-Tocopherol (IU/g)	0.045	10.045
Energy (kcal/g)	3.11	3.08
Crude Protein (%)	18.0	18.0
Crude Fat (%)	5.0	5.0
Crude Fiber (%)	5.0	5.0
Ash (%)	8.0	8.0

Abbreviations. CON: control chow; VIT: vitamin E-enhanced chow.

## Data Availability

The original data presented in the study are included in the article. Further inquiries can be directed to the corresponding authors.

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
