# Peer review of "Effects of Vitamin E Intake and Voluntary Wheel Running on Whole-Body and Skeletal Muscle Metabolism in Ovariectomized Mice"

_nutrients, 2025, doi:10.3390/nu17060991_

Round 1
Reviewer 1 Report
Comments and Suggestions for Authors
The manuscript by Jin et al., titled: "Effects of vitamin E intake and voluntary wheel running on whole-body and skeletal muscle metabolism in ovariectomized mice" is an interesting in vivo study aiming to investigate the effects of vitamin E and voluntary activity in overiectomized mice.
The manuscript is well structured and written. The reviewer would like to offer a few points for improvement below:
- Consider discussing the rationale for sample size (power calculation etc).
- Were the mice blocked prior to treatment/group assignment?
- How did the authors normalize for the differences in feed intake they observed?
- Did they also consider weights and normalized accordingly?
- It would be interesting to consider in the discussion more strongly the element of the microbiome.
English is fine proofreading is suggested.
Author Response
Thank you very much for taking the time to review this manuscript. Please find the detailed responses below. The corresponding revisions highlighted in red in the re-submitted files. Thank you.
Comments 1: Consider discussing the rationale for sample size (power calculation etc).
- Thanks a lot for your suggestions on sample size. A priori power analysis was conducted based on the previous research investigating the vitamin E effects on metabolism in ovariectomized rodents, and we used G*Power software (version 3.1.9.4) to calculate appropriate sample size (Azman et al. 2001, Ref#49). The analysis confirmed that 8 mice per group were minimally required in a between-subjects design (power = 0.00 and alpha = 0.05).
Comments 2: Were the mice blocked prior to treatment/group assignment?
- At least 24 hours prior to the experiment, we blocked the mice from using a voluntary running wheel. Thus, our metabolic data did not include an acute effect of exercise but a chronic effect of 12-week exercise treatment. We have clarified this by adding sentences in the 2.2 Experimental design.
Comments 3: How did the authors normalize for the differences in feed intake they observed?
- The vitamin E intake group was fed a diet containing vitamin E (10 IU/g, 1% weight of vitamin E to standard chow) for 13 weeks. The vitamin E group had free access to the vitamin E-supplemented chow. Please see the Table 1 dietary composition. Also, when we analyzed the food intake data, we used the averaged food intake per day in grams. We did not normalize the food intake amount to their body weight, since we thought that a significant group difference in body weight and fat mass may disrupt the actual group difference in food intake.
Comments 4: Did they also consider weights and normalized accordingly?
- Our animals body weights did not differ between groups when we assigned animals to groups at the beginning of the study. Thus, we thought that no further normalization was necessary when we compare the groups at the end.
Comments 5: It would be interesting to consider in the discussion more strongly the element of the microbiome.
- Thanks a lot for your recommendation and we totally agree on that. However, this manuscript has been focused on investigating the association between whole-body and skeletal muscle metabolism. If we put the element of the microbiome into our story, we thought that this could distract our story and we decided not to put those data in or mention previous findings in the discussion. Actually, we have prepared another manuscript which focuses on microbiome data and will be submitted to Nutrients soon.
Reviewer 2 Report
Comments and Suggestions for Authors
The manuscript presents an interesting study devoted to the effect of exercise and vitamin E supplementation on the whole-body metabolism, mass lean and fat mass accumulation, energy expenditure, glucose tolerance, PGC1-a, and AMPK levels and also protein carbonyl level in the muscle in ovariectomized mice. Ovariectomized mice could be a model for menopause. A somewhat unexpected though interesting finding was the highest body mass and fat mass gain in the exercising group supplemented with vitamin E.
I wonder whether 8 weeks was an optimal age for ovariectomy as a model for menopause.
Was the food intake monitored?
Was the exercise monitored?
Minor remarks:
Lines 43-44: The conclusions concerning post-menopausal women go beyond the results obtained. The authors are right but this statement shuld remain within Discussion.
Line 43: “ vitamin E would be a great source of antioxidants”, vitamin E is an antioxidant itself
Line 71: “Women live one-third of their lives after menopause”, one-third or more, let’s be optimistic
Line 210: please change “ml” to “mL”
Section 2.8: What about secondary antibodies (source, dilution and incubation conditions)?
Lines38, 385: “PGC”, “AMPK”, please explain the acronym at the first use
Author Response
Thank you very much for taking the time to review this manuscript. Please find the detailed responses below. The corresponding revisions highlighted in red in the re-submitted files. Thank you.
Comments 1: I wonder whether 8 weeks was an optimal age for ovariectomy as a model for menopause.
- Thanks a lot for your great comments on the optimal age for ovariectomy(OVX). Researchers studying the mechanisms of human menopause often use young, adult rodents (e.g., 8-12 weeks old) to establish the effects of long-term estrogen deficiency (PMID: 39408262, 34276568, 39415843). It is hard to say that OVX at those age ranges is optimal to mimic the menopause-associated metabolic syndrome in human. But, the OVX rodent model of ovarian suppression is innovative because it allows us to investigate and isolate the role of ovarian hormones independent of chronological aging. The complex hormonal changes during the menopausal transition make it difficult to isolate the effects of a single cause of metabolic syndrome in women in longitudinal studies. Using young, adult rodents at 8-12 weeks old is definitely beneficial in our research to investigate the role of ovarian suppression independent of chronological aging.
Comments 2: Was the food intake monitored?
- Yes, the food intake was weekly measured, and the data was analyzed and presented in three periods (early, mid, and late) by showing the averaged food intake during those periods. Please find the information at 2. Experimental design and 3.3. Food intake and wheel running distance.
Comments 3: Was the exercise monitored?
- Yes, the exercise distance was weekly monitored. Please find the information at 2. Experimental design and 3.3. Food intake and wheel running distance.
Comments 4: The conclusions concerning post-menopausal women go beyond the results obtained. The authors are right but this statement should remain within Discussion (Lines 43-44).
- Thank you for your comment. We acknowledge your concern and agree that the statement regarding post-menopausal women extends beyond our results obtained. To maintain clarity and ensure alignment with our findings, we have revised the Abstract and Conclusion by eliminating contents regarding menopausal women.
Comments 5: “ vitamin E would be a great source of antioxidants”, vitamin E is an antioxidant itself (line 43).
- Thank you for your comment. We acknowledge that vitamin E is itself an antioxidant. To enhance clarity and scientific accuracy, we have revised the statement in the Abstract.
Comments 6: “Women live one-third of their lives after menopause”, one-third or more, let’s be optimistic (line 71).
- Thank you for your thoughtful suggestion. We have revised the sentence to acknowledge that women spend one-third or more of their lives after menopause, providing a more optimistic perspective to readers. Please see the line 73.
Comments 7: Please change “ml” to “mL” (lines 210).
- We fixed ml to mL throughout the manuscript. Thank you (line 163, 202, and 212).
Comments 8: What about secondary antibodies (source, dilution and incubation conditions)? (Section 2.8).
- Thank you for your careful review. We have now provided the information of secondary antibody. Please see 8. Western blot analysis.
Comments 9: “PGC”, “AMPK”, please explain the acronym at the first use (lines 38).
- Thank you for your careful review. We have now provided the full names of “PGC” and “AMPK” at their first mention to ensure clarity for all readers. Please see the abstract and 8. Western blot analysis.
Reviewer 3 Report
Comments and Suggestions for Authors
Specific: I think your conclusion should be tempered. It seems like a leap to far to say vitamin E would be good for women when you have only tested on mice.
General
L43: delete "in conclusion" (its repetitive)
L51: what is the biggest physiological change? This sentence seems like it could be improved for presentation/grammar
L62: probably due to formatting, but look to decrease spacing 60%
L66: hyperneutrophilia
L127: avoid 1st person language
L133: Forty of? Is "of" a mistake?
L167-171: spacing between # and %
L167: avoid 1st person language
L219: three
L245: assume normality? Was normality tested? If so, report
Figures and legends look appropriate
L395, 413, 428, 514, 521: avoid 1st person language
Did I miss figure legends and callouts (Figure 1) in the text?
Author Response
Thank you very much for taking the time to review this manuscript. Please find the detailed responses below. The corresponding revisions highlighted in red in the re-submitted files. Thank you.
Comments 1: I think your conclusion should be tempered. It seems like a leap to far to say vitamin E would be good for women when you have only tested on mice.
- Thank a lot for your great comments. We acknowledge your concern and agree that the statement regarding post-menopausal women extends beyond our results obtained. To maintain clarity and ensure alignment with our findings, we have revised the Abstract and Conclusion by eliminating contents regarding menopausal women.
Comments 2: delete "in conclusion" (its repetitive) (line 43).
- We fixed the sentence in the abstract. Thank you.
Comments 3: what is the biggest physiological change? This sentence seems like it could be improved for presentation/grammar (line 51).
- Thank you for your comment. We acknowledge your concern and we revised the sentence accordingly (line 52-53).
Comments 4: Probably due to formatting, but look to decrease spacing 60% (line 62).
- We fixed it. Thank you.
Comments 5: Hyperneutrophilia (line 66).
- We fixed the word. Thank you.
Comments 6: Avoid 1st person language (line 127).
- We changed the sentence to “The hypothesis was that vitamin E intake combined with exercise treatment would favorably affect whole-body metabolism (i.e., fat mass, energy expenditure, glucose tolerance) and alleviate skeletal muscle mitochondrial dysfunction in ovariectomized mice”. Thank you.
Comments 7: Forty of? Is "of" a mistake? (line 133).
- Sorry for the grammar mistake. We have removed the word “of”. We appreciate your careful review.
Comments 8: Spacing between # and % (lines 167-171).
- We unified the format of spacing between number and % throughout the manuscript.
Comments 9: Avoid 1st person language (line 167).
- We fixed the sentence. Thank you.
Comments 10: Three (line 219).
- We fixed the number to a word. Thank you.
Comments 11: Assume normality? Was normality tested? If so, report (line 245).
- We totally agree on your suggestion. Normality tests such as the Shapiro-Wilk test should be performed before running ANOVAs. However, one of the statistical limitations of pre-clinical research using rodents is that each experimental group has very minimal sample size due to ethical and cost issues. In the current study, we assumed their normality, and it is highly possible that some of the variables did not pass the criteria of normality since we have only 8 animals per group. We put this in the potential research limitations (line 529-531). Thanks a lot for your great suggestions.
Comments 12: Figures and legends look appropriate.
- Thank you for the positive comment.
Comments 13: Avoid 1st person language (lines 395, 413, 428, 514, and 521).
- We revised sentences throughout the manuscript. Thank you.
Comments 14: Did I miss figure legends and callouts (Figure 1) in the text?
- Sorry for the confusion. We put the body weight, fat and lean mass data in Figure 1 with the legends. For a better presentation, we switched the section order of 2. Food intake and wheel running distance and 3.3. Fat and lean mass. The text of body weight is now located in 3.1. Body weight, and the fat and lean mass are in 3.2. Fat and lean mass. Thank you.